



# Measurement Error Proxy System Models: MEPSM v0.2

Matt J. Fischer[1]

[1]NST Environment, ANSTO

**Correspondence:** Matt Fischer (mjf@ansto.gov.au)

**Abstract.** Proxy system models (PSMs) are an essential component of paleoclimate data assimilation and for testing climate field reconstruction methods. Generally, current statistical PSMs consider the noise in the output (proxy) variable only, and ignore the noise in the input (environmental) variables. This problem is exacerbated when there are several input variables. Here we develop a new PSM, the Measurement Error Proxy System Model (MEPSM), which includes noise in all variables, including noise auto- and cross-correlation. The MEPSM is calibrated using a quasi-Bayesian solution, which leverages Gaussian conjugacy to produce a fast solution. Another advantage of MEPSM is that the prior can be used to stabilize the solution between an informative prior (e.g. with a non-zero mean) and the maximum likelihood solution. MEPSM is illustrated by calibrating a proxy model for $\delta^{18}O_{coral}$ with multiple inputs (marine temperature and salinity), including noise in all variables. MEPSM is applicable to many different climate proxies, and will improve our understanding of the effects of predictor noise on PSMs, data assimilation, and climate reconstruction.

## 1 Introduction

Proxy system models (PSMs) describe how biological, geological or chemical archives are imprinted with environmental signals (Evans et al., 2013; Dee, 2015). They are an essential component of paleoclimate data assimilation (Steiger et al., 2018; Tardif et al., 2019; Sanchez et al., 2021; King et al., 2023), and of pseudoproxy experiments e.g., testing the fidelity of climate reconstruction methods (Loope et al., 2020a, b). For PSMs, input variables include one or more environmental variables (e.g. temperature, water salinity, rainfall), and output variables include variables which can be read from natural archives e.g. the abundance of trace elements or isotopes in carbonate archives. The processes between input and outputs may be linear or nonlinear fitted relationships (statistical PSMs), or more detailed physiochemical models (physiochemical PSMs). Statistical PSMs can be frequentist (i.e. the PSM is calibrated using ordinary least squares, OLS) or Bayesian (PSMs that make use of Bayes' Rule). Current paleoclimate data assimilation projects often use frequentist PSMs (e.g. Steiger et al., 2018; Tardif et al., 2019) (see also example 1 in King et al., 2023), but some paleodata assimilation projects have begun to incorporate Bayesian PSMs (King et al., 2023). Current Bayesian PSMs (e.g. Tierney and Tingley, 2014, 2018; Tierney et al., 2019; Malevich et al., 2019) typically have the form:

$$\zeta \sim \mathcal{N}(b_0 + f(\boldsymbol{x}, \Theta), \ \sigma_\zeta^2) \tag{1}$$

that is, $\zeta$ is sampled from the Normal distribution $\mathcal{N}(\mu, \sigma^2)$, where $b_0$ is the regression intercept, $\boldsymbol{x}$ and $\zeta$ are the input and output variables respectively (note $\boldsymbol{x}$ is a row vector of predictor matrix $\mathbf{X}$), and $\Theta$ is a set of parameters. These Bayesian



models are based on Bayesian OLS rather than Bayesian TLS (total least squares). In comparison, an errors-in-variables (EIV) model (containing error in both $\boldsymbol{x}$ and $y$) may have the form:

$$y \sim \mathcal{N}(b_0 + f(\boldsymbol{x}, \boldsymbol{B}),\ df(\boldsymbol{x}, \boldsymbol{B})'\Sigma_{aa}df(\boldsymbol{x}, \boldsymbol{B})\ +\ \varepsilon_y^2) \tag{2}$$

where $df(\boldsymbol{x}, \boldsymbol{B})$ is the derivative of $f(\boldsymbol{x}, \boldsymbol{B})$ for each observation vector $[y\ \boldsymbol{x}]$, the prime symbol $(')$ is the tranpose operator, $\Sigma_{aa}$ is the covariance matrix of the noise associated with $\boldsymbol{x} = \boldsymbol{x}^* + \boldsymbol{a}$, and $\boldsymbol{x}^*$ denotes the part of $\boldsymbol{x}$ that is unobservable. Appendix 1 contains further information about Eq. (2). The term measurement error is used here generally to describe any uncertainty associated with the value of the predictors and the response variable, which could include both measurement uncertainty (e.g. from a thermometer), as well as methodological uncertainty (from the method used to "combine" several measurements, which

may be spatially and temporally apart). The model in Eq. (2) is not identifiable, meaning the parameters cannot be uniquely determined without additional information, such as information about $\Sigma_{aa}$. A full Bayesian solution to the problem may also require reformulating the model in terms of a subspace of $[\boldsymbol{Y}\ \ \boldsymbol{X}]$, see e.g., Florens et al. (1974). The focus of this paper is on a quasi-Bayesian solution to Eq. (2), which leverages Gaussian conjugacy instead of a Markov Chain Monte Carlo (MCMC) solution.

In this paper, EIV and WLS (weighted least squares) refer to general approaches. York (1966, 1968) introduced a root-finding solution for one-predictor WLS regression. Ludwig and Titterington (1994) presented a maximum likelihood (ML) solution for a straight line in 3-dimensional space, with heterogenous noise in all variables. Schneider (2001) briefly discussed the multivariate method of total least squares, which assumes homogenous and identically-distributed noise for both predictor and response fields. Hannart et al. (2014) introduced a ML algorithm for weighted TLS (WTLS), that accounts for heterogeneity

and temporal autocorrelation in the predictor noise and the response noise, but lacks noise cross-correlation. Also ML solutions, by their very nature, don't incorporate prior information about the regression coefficients. This study exapands the model of Hannart et al. (2014), to include both prior information and generalized noise (auto- and cross-correlation within and between the noise series), in a Measurement Error Proxy System Model (MEPSM). The MEPSM is formulated in Sect. 2.1, and steps for practical implementation are given in Sect. 2.2. MEPSM is applied to a real example in Sect. 3.





**Table 1.** Notation used.

| Symbol | Definition |
|--------|-----------|
| $\mathbf{A}$ | $n \times p$ matrix of predictor noise, $p$ predictors with $n$ time points |
| $\boldsymbol{B}$ | $(p+1)$-length vector of regression weights |
| $\boldsymbol{e}$ | $n$-length vector of noise, $\boldsymbol{Y} - \mathbf{X}\boldsymbol{B}$ |
| $\mathbb{I}_d$ | Identity matrix of rank $d$ |
| $\mathbf{Q}$ | $n \times (p+1)$ matrix of noise for all variables, $\mathbf{Q} = [\boldsymbol{V} \quad \mathbf{A}]$ |
| $\mathbf{R}$ | $n \times n$ auto- or cross-correlation matrix |
| $\Sigma$ | Covariance matrix |
| $\boldsymbol{V}$ | $n$-length vector of response noise, $\boldsymbol{V} = \boldsymbol{Y} - \boldsymbol{Y}^*$ |
| $\mathbf{W}$ | $n \times n$ weight matrix |
| $\mathbf{X}$ | $n \times (p \text{ or } p+1)$ matrix, $p$ predictors with $n$ time points, may include an intercept term |
| $\boldsymbol{Y}$ | $n$-length response vector |
| $\mathbf{Z}$ | $n \times (p+1)$ matrix, $\mathbf{Z} = [\boldsymbol{Y} \quad \mathbf{X}]$ |
| $\Omega$ | $n \times n$ lag-covariance matrix of noise |

## 2 Methods

### 2.1 MEPSM

The general EIV model can be expressed as:

$$\boldsymbol{z}_t = \boldsymbol{z}_t^* + \boldsymbol{q}_t \tag{3a}$$

$$\boldsymbol{z}_t^* = [y_t^* \quad \boldsymbol{x}_t^*] \tag{3b}$$

$$\boldsymbol{q}_t = [v_t \quad \boldsymbol{a}_t] \tag{3c}$$

$$y_t^* = b_0 + \boldsymbol{x}_t^* \boldsymbol{B}^* + \epsilon_t \tag{3d}$$

where $\boldsymbol{z}_t$ are the observed data, $\boldsymbol{z}_t^*$ are unobserved values underlying $\boldsymbol{z}_t$, $\boldsymbol{q}_t$ is a row vector of noise matrix $\mathbf{Q} = [\boldsymbol{V} \quad \mathbf{A}]$, $\boldsymbol{B}^*$ is a vector of regression coefficients, and $\epsilon_t$ is the equation error. Note that here the superscript star has two uses: generally it denotes the part of a variable which is unobservable, but for $\boldsymbol{x}^*$ and $\boldsymbol{B}^*$ it also denotes vectors (or arrays) without an intercept term. Further, vector $\boldsymbol{B}$ may be thought of as unobserved because it is calculated not measured, so the main difference between $\boldsymbol{B}$ and $\boldsymbol{B}^*$ is the exclusion of an intercept coefficient in $\boldsymbol{B}^*$ i.e. $\boldsymbol{B} = [b_0 \quad \boldsymbol{B}^*]'$. From Eq. (3), the relevant covariance matrices are:

$$\Sigma_{z^*z^*} = \begin{bmatrix} \sigma_{y^*y^*}^2 & \Sigma_{y^*X^*} \\ \Sigma_{X^*y^*} & \Sigma_{X^*X^*} \end{bmatrix}, \qquad \Sigma_Q = \begin{bmatrix} \Omega_V & \mathbf{0} \\ \mathbf{0} & \Sigma_A \end{bmatrix}, \quad \text{and} \tag{4}$$





$$\Sigma_A = \begin{bmatrix} \Omega_{A_1} & \Omega_{A_1 A_2} & \dots & \Omega_{A_1 A_p} \\ \Omega_{A_2 A_1} & \Omega_{A_2} & \dots & \Omega_{A_2 A_p} \\ \vdots & \vdots & \ddots & \vdots \\ \Omega_{A_p A_1} & \Omega_{A_p A_2} & \dots & \Omega_{A_p} \end{bmatrix}, \tag{5}$$

where e.g. $\Sigma_Q$ means the covariance matrix of $\mathrm{vec}(\mathbf{Q})$, while $\Sigma_{QQ}$ means the covariance matrix of $\mathbf{Q}$. The operation $\mathrm{vec}(\mathbf{Q})$ means to column stack the matrix $\mathbf{Q}$ (Henderson and Searle, 1979). The matrix $\Sigma_{z^* z^*}$ is assumed to be homogenous in time. The matrices $\Omega$ are $n \times n$ matrices that describe e.g. the temporal autocovariance of the error variable $\mathbf{A}_i$ ($\Omega_{A_i}$), and cross-covariance with the error variable $\mathbf{A}_j$ ($\Omega_{A_i A_j}$). Unlike the algorithm in Hannart et al. (2014), I do not assume that the off-diagonal $\Omega_{A_i A_j}$ are $\mathbf{0}_{n \times n}$ matrices. Also note that in this paper $\Omega_V = \Sigma_V$. More will be said about $\Sigma_Q$ in Sect. 3.1.

In solving the general EIV model, the equation error $\epsilon_t$ (Eq. 3d) is often ignored as being separate from $v_t$. An exception is in Fuller (1987) in which $\epsilon_t$ is estimated by keeping the predictor noise within $p \times p$ matrices $\Sigma_{a_t a_t}$ (which is like a special case of $\Sigma_A$). I also ignore the equation error, so the model in Eq. 3 can be re-expressed:

$$\begin{bmatrix} Z_{11} \\ Z_{21} \\ \vdots \\ Z_{n1} \end{bmatrix} = \begin{bmatrix} 1 & \boldsymbol{x}_1 \\ 1 & \boldsymbol{x}_2 \\ \vdots \\ 1 & \boldsymbol{x}_n \end{bmatrix} \begin{bmatrix} b_0 \\ \boldsymbol{B}^* \end{bmatrix} + \begin{bmatrix} \mathbb{I}_n & -(\boldsymbol{B}^{*\prime} \otimes \mathbb{I}_n) \end{bmatrix} \begin{bmatrix} \boldsymbol{V} \\ \mathrm{vec}(\mathbf{A}) \end{bmatrix} \tag{6}$$

where $\boldsymbol{V} \sim \mathcal{N}(\mathbf{0}_n, \, \Omega_V)$, $\mathrm{vec}(\mathbf{A}) \sim \mathcal{N}(\mathbf{0}_{n \cdot p}, \, \Sigma_A)$, and here $\boldsymbol{x}_t = \boldsymbol{x}_t^* + \boldsymbol{a}_t$. The response variable is the first column of $\mathbf{Z}$,

denoted as $\mathbf{Z}_1$ or more generally as $\boldsymbol{Y}$. Note also the two possible definitions of response noise for the model: $\boldsymbol{V} = \boldsymbol{Y} - b_0 - \mathbf{X}^* \boldsymbol{B}^*$ and $\boldsymbol{e} = \boldsymbol{Y} - \mathbf{X} \boldsymbol{B}$ (where $\mathbf{X}$ includes an intercept term). The difference is that:

$$\boldsymbol{V} = \boldsymbol{Y} - \mathbf{X} \boldsymbol{B} + \mathbf{A} \boldsymbol{B}^* = \boldsymbol{e} + \mathbf{A} \boldsymbol{B}^* \tag{7}$$

(again $\mathbf{X}$ includes a column of ones for the regression intercept).

     From $\boldsymbol{e} = \boldsymbol{V} - \mathbf{A} \boldsymbol{B}^*$, the covariance matrix of model prediction can be expressed as:

$$\mathbf{W} = \Omega_V + (\boldsymbol{B}^{*\prime} \otimes \mathbb{I}_n) \Sigma_A (\boldsymbol{B}^* \otimes \mathbb{I}_n) \tag{8}$$

     Different solutions to the general model exist, including a Generalized Least Squares (GLS) estimator (Fuller, 1990), Weighted Total Least Squares (WTLS) (Amiri-Simkooei and Jazaeri, 2012), and a maximum a posterior (MAP) estimator (Fang, 2017). The latter method is partly based on GLS, because it uses the formulation (in my notation):

$$\boldsymbol{B}_{\mathrm{gls}} = \left( \mathbf{X}^{*\prime} \Omega_V^{-1} \mathbf{X}^* \right)^{-1} \left( \mathbf{X}^{*\prime} \Omega_V^{-1} \boldsymbol{Y} \right), \tag{9}$$

for the maximum likelihood component. In these studies, a current issue is the different ways of calculating $\Sigma_B$. For example, Amiri-Simkooei and Jazaeri (2012) and Fang (2017) give the covariance matrix of $\Sigma_B$ as (in my notation): $\Sigma_B =$





$\sigma^2 \left( \mathbf{X}^{*'} \, \mathbf{W}^{-1} \, \mathbf{X}^* \right)^{-1}$ where $\sigma^2 = \frac{e' \, \mathbf{W}^{-1} \, e}{n-p}$, and $\Sigma_B = \left( \mathbf{X}^{*'} \, \Omega_V^{-1} \, \mathbf{X}^* \right)^{-1}$, respectively without theoretical derivation. See Appendix B for further information on covariance notation.

Bayes' rule states that the posterior density function $p(\boldsymbol{\Theta}|\mathcal{D})$ is proportional to the product of the likelihood function $L(\boldsymbol{\Theta})$
and the prior density function $p(\boldsymbol{\Theta})$:

$$p(\boldsymbol{\Theta}|\mathcal{D}) \;\propto\; L(\boldsymbol{\Theta}) \times p(\boldsymbol{\Theta}) \tag{10}$$

Note that it does not matter if the likelihood function is a scaled distribution e.g., a normal density function multiplied by a scalar (so it integrates to a number other than unity).

Assuming that the right-side distributions are Gaussian (or scaled Gaussian), then

$$\mathcal{N}(\boldsymbol{B}, \, \Sigma_B) \;\propto\; \mathcal{N}(\boldsymbol{B}_{\mathrm{ml}}, \, \Sigma_B^{\mathrm{ml}}) \times \mathcal{N}(\boldsymbol{B}_0, \, \Sigma_0) \tag{11}$$

where ml refers to the maximum likelihood solution. The posterior $\mathcal{N}(\boldsymbol{B}, \, \Sigma_B)$ integrates to unity, as expected. First I show an example by using the GLS likelihood. Let the GLS likelihood be:

$$\mathcal{N}(\boldsymbol{B}_{\mathrm{ml}}, \, \Sigma_B^{\mathrm{ml}}) \;=\; \mathcal{N}\!\left( (\mathbf{X}' \Sigma^{-1} \mathbf{X})^{-1} \mathbf{X}' \Sigma^{-1} \boldsymbol{Y}, \, (\mathbf{X}' \Sigma^{-1} \mathbf{X})^{-1} \right) \tag{12}$$

Now apply the gaussian multiplication identity:

$$\mathcal{N}(\mathbf{C}(\Sigma_B^{\mathrm{ml}\,-1} \boldsymbol{B}_{\mathrm{ml}} + \Sigma_0^{-1} \boldsymbol{B}_0), \, \mathbf{C}) \;\propto\; \mathcal{N}(\boldsymbol{B}_{\mathrm{ml}}, \, \Sigma_B^{\mathrm{ml}}) \times \mathcal{N}(\boldsymbol{B}_0, \, \Sigma_0) \tag{13}$$

where $\mathbf{C} = \left( \Sigma_B^{\mathrm{ml}\,-1} + \Sigma_0^{-1} \right)^{-1}$. Thus the mean and varaince of $p(\boldsymbol{\Theta}|\mathcal{D})$ are:

$$\boldsymbol{B} \;=\; (\mathbf{X}' \Sigma^{-1} \mathbf{X} + \Sigma_0^{-1})^{-1} (\mathbf{X}' \Sigma^{-1} \boldsymbol{Y} + \Sigma_0^{-1} \boldsymbol{B}_0) \tag{14a}$$
$$\Sigma_B \;=\; (\mathbf{X}' \Sigma^{-1} \mathbf{X} + \Sigma_0^{-1})^{-1} \tag{14b}$$

The above solution is a simple example using the GLS likelihood. To obtain the solution for the EIV problem, the GLS
likelihood is replaced with the WTLS likelihood $\mathcal{N}(B_{\mathrm{wtls}}, \, \Sigma_B^{\mathrm{wtls}})$, where $\boldsymbol{B}_{\mathrm{wtls}}$ and $\Sigma_B^{\mathrm{wtls}}$ are derived in Appendix C and D. Using the WTLS likelihood, simplifying the product $\Sigma_B^{\mathrm{ml}\,-1} \boldsymbol{B}_{\mathrm{ml}}$ (from Eq. 13) is no longer feasible, so instead we write the mean and variance of $p(\boldsymbol{\Theta}|\mathcal{D})$ simply as:

$$\boldsymbol{B} \;=\; \left( \Sigma_B^{\mathrm{wtls}\,-1} + \Sigma_0^{-1} \right)^{-1} \left( \Sigma_B^{\mathrm{wtls}\,-1} \boldsymbol{B}_{\mathrm{wtls}} + \Sigma_0^{-1} \boldsymbol{B}_0 \right) \tag{15}$$
$$\Sigma_B \;=\; \left( \Sigma_B^{\mathrm{wtls}\,-1} + \Sigma_0^{-1} \right)^{-1}. \tag{16}$$

## 110 2.2 Implementation

An application to calibrate a real PSM is provided in Sect. 3. Here general aspects of the solution and implementation are discussed.

The predictor noise and response noise, $\mathbf{A}$ and $\boldsymbol{V}$, can be estimated as:

$$\mathbf{A} \;=\; -(\boldsymbol{B}^{*'} \otimes \mathbb{I}_n) \Sigma_A (\mathbb{I}_p \otimes \mathbf{W}^{-1} \boldsymbol{e}) \tag{17}$$





$$V = \Omega_V \mathbf{W}^{-1} e \tag{18}$$

where $e = Y - \mathbf{X}B$. Matrix $\mathbf{A}$ is needed for the calculation of both $B_{\text{wtls}}$ and $\Sigma_B^{\text{wtls}}$, while $V$ is needed to reestimate $\Omega_V$ if relevant (see next paragraph).

The $n \times n$ covariances matrices $\Omega_{A_i}$ (or $\Omega_{A_i A_j}$) and $\Omega_V$, ideally need information about the cross-correlation within $\mathbf{A}$, and autocorrelation of $\mathbf{A}$ and $V$. If the ACF (autocorrelation function) and CCF (cross-correlation function) are not available, these can potentially be estimated using an iterative procedure. The procedure begins with vectors $\sigma_A^2$ and $\sigma_V^2$, which are the heterescedastic error variances (the diagonals of $\Sigma_A$ and $\Omega_V$).

1. Initialize $\Sigma_A = \text{Diag}(\sigma_A \odot \sigma_A)$ and $\Omega_V = \text{Diag}(\sigma_V \odot \sigma_V)$, where $\odot$ denotes Hadamard multiplication

2. Estimate the first posterior: $\mathcal{N}(B, \Sigma_B)$, and the predictor and response noise: $\mathbf{A}$ and $V$ (Eqs. 15-18)

3. Calculate the ACF (for $\mathbf{A}$ and $V$) and the CCF (for $\mathbf{A}$)

4. Estimate $\Sigma_A$ and $\Omega_V$ as in Appendix E, and recalculate the posterior distribution.

## 3 Application

### 3.1 A proxy system model, uncertainties, and the prior

A bivariate (i.e. two-predictor) PSM for coral $\delta^{18}$O is:

$$\delta^{18}\text{O}_{\text{coral}} = b_0 + b_1 \text{SST} + b_2 \text{SSS} \tag{19}$$

where SST is sea surface temperature and SSS is sea surface salinity. Note that in the coral literature, the coefficients $b_1$ and $b_2$ are commonly referred to as $a_1$ and $a_2$, but here I use the former symbols for generality. This bivariate PSM has been used in data assimilation (Sanchez et al., 2021) and Monte Carlo experiments (Thompson et al., 2022). The bivariate PSM can be extended by rewriting it in the form of Eq. (6). The MEPSM requires estimates of the time-varying uncertainty of all variables, and a prior distribution for $B$ (see below). Here the time-varying uncertainties for the predictors are obtained from recent SST and SSS products, that provide the complete uncertainty field $\varepsilon^2(x, y, t)$ i.e. the uncertainty variance for each grid cell, at each time point.

For ERSSTv5, for each grid cell and time point, the SST total uncertainty $\varepsilon_T^2(x, y, t)$ is the sum of the parametric uncertainty and reconstruction uncertainty. These uncertainties are derived from a large reconstruction ensemble (Huang et al., 2020). The parametric uncertainty is based on the difference between the ensemble average and each ensemble member, given various internal parameters which affect SST uncertainty (including the uncertainty of the ship/buoy/float measurement). The reconstruction uncertainty is based on the difference between each ensemble member and a pseudo-observation dataset on which the reconstruction method is trained e.g. OISST (Optimally Interpolated SST). Note that the calculation of reconstruction uncertainty includes grid cells that have observational or no observational data, while the calculation of parametric uncertainty includes only grid cells that have observational data.



For HadEN4, the SSS total uncertainty $\varepsilon_S^2(x,y,t)$ is a combination of background uncertainty and observational uncertainty (Good et al., 2013). The observational uncertainty is obtained from previous studies, while the background uncertainty is the uncertainty associated with a persistent-based forecast.

For $\delta^{18}O_{coral}$, the main known uncertainty is the analytical uncertainty, which is $\sim 0.05$ ‰ for $\delta^{18}O_{coral}$ (Osborne et al., 2013). Time variation could be included by expressing the analytical uncertainty as a relative standard deviation of the mean $\delta^{18}O$ (the effect of this will be tested elsewhere). Other uncertainities, which are unknown at many sites, include the intra- and inter-colony noise (Sayani et al., 2019). Note also that detrending adds time-varying uncertainty, because the trend uncertainty is larger at the ends of the time series (this is applicable to all detrended variables).

For MEPSM, the matrix $\Sigma_A$ can be initially constructed using $\text{Diag}(\sigma_A \odot \sigma_A)$ where $\sigma_A' = [\varepsilon_T(site,\cdot), \varepsilon_S(site,\cdot)]$, $site$ refers to the grid cell of a particular site, and the $(site,\cdot)$ notation means over all time points. In the SST and SSS products, no information is given on the noise serial correlation, and no information exists on the noise cross-correlation between SST and SSS products. Seasonal peaks in the autocorrelation function (ACF) of SST noise could arise by e.g. seasonal changes in shipping tracks, which would affect the parametric uncertainty. Cross-correlation between SST and SSS noise might arise from sampling these variables at the same points within a particular grid cell (i.e. from the same ship), or due to seasonal variation in shipping tracks. Future work should investigate the ACF and CCF (cross-correlation function) for the SST and SSS ensembles (this is beyond the scope of the current paper). Finally, since the uncertainties in the predictors and response are fundamentally different, then $\Sigma_Q$ is assumed to be a block-diagonal matrix, with $\Omega_V$ and $\Sigma_A$ on the block diagonal, and zero-filled blocks on the block anti-diagonal (Eq. (4)).

For Bayesian methods, informative prior distributions are thought to be more useful than noninformative priors (Lemoine, 2019) e.g., when using a fully noninformative prior distribution the posterior distribution will be close to the maximum likelihood distribution, which would make using the fully noninformative prior somewhat obselete.

The prior distribution for $\boldsymbol{B}$ is obtained from previous values that were used in Monte Carlo PSM experiments (Thompson et al., 2022; Watanbe and Pfeiffer, 2022). Those previous studies did not seek to calculate Bayesian posterior distributions, but the values in those studies represent current beliefs about the coefficients and their uncertainty, and can be used as a prior. The prior distribution obtained from those studies is:

$$p(B) = \mathcal{N}(\boldsymbol{B}_0, \Sigma_0) = \mathcal{N}\left(\begin{bmatrix} b_0 \\ -0.22 \\ 0.27s \end{bmatrix}, \begin{bmatrix} \sigma_{b_0}^2 & 0 & 0 \\ 0 & 0.02^2 & 0 \\ 0 & 0 & 0.15^2 \end{bmatrix}\right) \tag{20}$$

where $s = 0.97$ is a scaling factor between $\delta^{18}O_{seawater}$ (in VSMOW) and $\delta^{18}O_{coral}$ (in VPDB). The values of $b_1 = -0.22$ and $\sigma_{b_1}^2 = 0.02^2$ are from Thompson et al. (2022) and Watanbe and Pfeiffer (2022) respectively. The value of $s^{-1}b_2 = 0.27$ ‰ psu$^{-1}$ originates from LeGrande and Schmidt (2006), and is an average ‰/salinity value for the Tropical Pacific Ocean. Regional variations in this value will probably be revealed by new datasets (DeLong et al., 2022), but for the purpose of this example of MEPSM, the value of $0.27$ ‰ psu$^{-1}$ is used. Thompson et al. (2022) considered two values for $\sigma_{b_2}^2$ i.e. $\sigma_{b_2}^2 = \{0.1^2, 0.2^2\}$, so for this example an "average" value is used ($0.15^2$). The value of the intercept $b_0$, and it's uncertainty




$\sigma_{b_0}^2$, are unknown in the sense that they are site-dependent values. A prior for these unknown values can be estimated by using the other prior coefficients and SST and SSS data for a particular site. For a particular site, the prior value of $b_0$ can be calculated as:

$$b_0 = \mu_y - u_X \begin{bmatrix} -0.22 \\ 0.27s \end{bmatrix} \tag{21}$$

where $\mu_X = [\mu_{X_1} \quad \mu_{X_2}]$ and $\mu_y$ are the mean values of the predictors and response. From ordinary least sqaures theory, the diagonal elements of the covariance of $\boldsymbol{B}$ are typically given as:

$$\sigma_{b_j}^2 = \sigma_e^2 / \mathrm{RSS}_j \tag{22}$$

where $\sigma_e^2$ is the error variance, and $\mathrm{RSS}_j$ is the residual sum of squares after regressing $\mathbf{X}_j$ on $\mathbf{X}_{-j}$, where $\mathbf{X}_{-j}$ is the matrix $\mathbf{X}$ omitting column $j$. So for $\sigma_{b_0}^2$:

$$\mathrm{RSS}_0 = n - n^2 \mu_X' (\mathbf{X}_{-0}' \mathbf{X}_{-0})^{-1} \mu_X \tag{23}$$

where $\mathbf{X}_{-0}$ is the observed part of $\mathbf{X}$ not including an intercept term (different than $\mathbf{X}^*$ which is the unobserved part of $\mathbf{X}$). Preliminary analysis showed that the prior value for $\sigma_{b0}^2$ had a large influence on the posterior distribution of $\boldsymbol{B}$, so $\sigma_{b_0}^2$ only was scaled to be uninformative i.e. $10^3 \sigma_{b_0}^2$.

## 3.2 Coral example

Rock Islands (7.27°N, 134.38°E) is a site on the western barrier reef of Palau, western Pacific Ocean. A number of coral cores were sampled from Rock Islands and surrounds, but the focus here is on one core labelled RI6, which is a monthly-resolved record spanning 1899-2008. The core was sampled from a narrow-entrance lagoon, in shallow water ($\sim$2 metres depth). Osborne et al. (2013) compared the coral $\delta^{18}$O record with temperature data from the NCEP-NCAR Reanalysis (Kalnay 2006), and salinity data from LEGOS (Delcroix 2011). Using the raw (not detrended or deseasonalized) RI record from 1970-2008, and ordinary least squares, Osborne et al. (2013) regressed the coral $\delta^{18}$O against surface temperature and salinity, producing the equation:

$$\delta^{18}\mathrm{O}_{\mathrm{coral}} = -0.06 \mathrm{SST} + 0.43 \mathrm{SSS} - 18.70; \tag{24}$$

see Supplementary Table A4 in Osborne et al. (2013). Osborne et al. attributed the small SST coefficient to the small seasonal SST range at Palau ($\sim$1.5°C), presumably because the small SST range makes the data scatter with $\delta^{18}$O more spherical than ellipsoid. They did not directly compare their raw SSS coefficient with any other data. Hence the RI $\delta^{18}$O record was chosen for this example, in order to further investigate the SST and SSS effects on coral $\delta^{18}$O at this site. In the following analysis the main differences with the analysis of Osborne et al. (2013) are that here the SST and SSS data are extracted from ERSSTv5 and HadEN4, because these products provide the uncertainty fields (Sect. 3.1). Secondly, the time period examined is from 1950-2008. Also, for this study, all variables were detrended (using linear detrending and without removing the intercept), in order



to ensure stationarity in the mean. For all variables, the trends at this site were weak e.g. for $\delta^{18}O_{coral}$ the trend from 1950-2008 was $-0.00135\pm0.004$ ‰ per year. A Jupyter notebook with these prepocessing steps is available in the Supplementary Material. Using this updated (and detrended) data, the OLS regression was recalculated as:

$$\delta^{18}O_{coral} = -0.13\text{SST} + 0.42\text{SSS} - 16.25. \tag{25}$$

Next MEPSM was applied to the same data, following the steps in Sect. 2.2. Except, for the application presented here, only $\Sigma_A$ was updated in step 4. $\Omega_V$ was set as $\Omega_V = \text{Diag}(\sigma_V \odot \sigma_V)$ throughout, because here $\sigma_V$ is thought to be mostly analytical noise, which should be approximately white (the uncertainty due to detrending is time-varying, but also white). For $\Sigma_A$, for the auto- and cross-correlation functions of the predictor noise, up to 100 lags (months) were retained to construct the submatrices $\mathbf{R}_{A_i}$ and $\mathbf{R}_{A_i A_j}$ (see Appendix E). Experiments with 50 and 150 lags showed little difference in the final posterior. In any application of MEPSM, the user should consider the source of the predictor and response noise in constructing $\Sigma_A$ and $\Omega_V$ (as in Sect. 3.1).

Figure 1 shows the prior and the first and final posterior distributions for the SST and SSS coefficients. The first posterior distribution (dashed line) is obtained after step 2 (Sect. 2.2), while the final posterior is obtained after updating $\Sigma_A$ for auto- and cross-correlation. The (final) mean SST and SSS coefficients are given in Figure 2: $b_1 = -0.25$ ‰ °C$^{-1}$ and $b_2 = 0.49$ ‰ psu$^{-1}$. Figure 2 shows the marginal plots of the 3-dimensional scatter plot (SST, SSS, $\delta^{18}O$), together with the prior and posterior regression lines. For Rock Island, the SST and SSS coefficients are both steeper for the final posterior, compared to the OLS equation (Eq. 25).





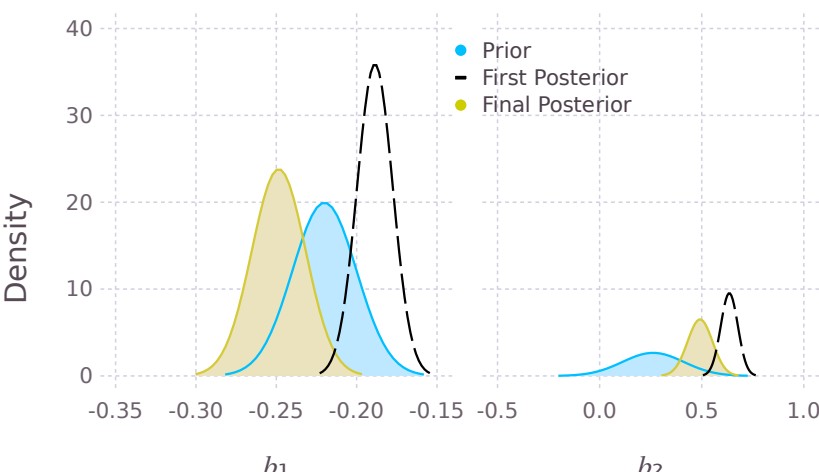

**Figure 1.** The prior and posterior distributions for the SST ($b_1$) and SSS ($b_2$) coefficients. The final posterior is wider than the first posterior, because the final posterior accounts for auto- and cross-correlation in $\Sigma_A$. In Figs. 1-3 the first posterior is the dashed line.

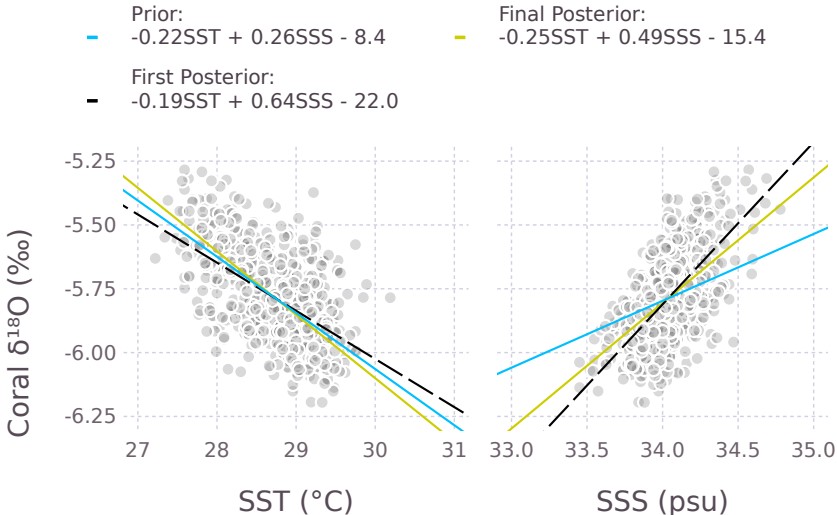

**Figure 2.** The regression lines of the prior and posterior distributions. In 3d, these regression lines are the sides of a plane. Each shaded point corresponds to a monthly value from 1950-2008.





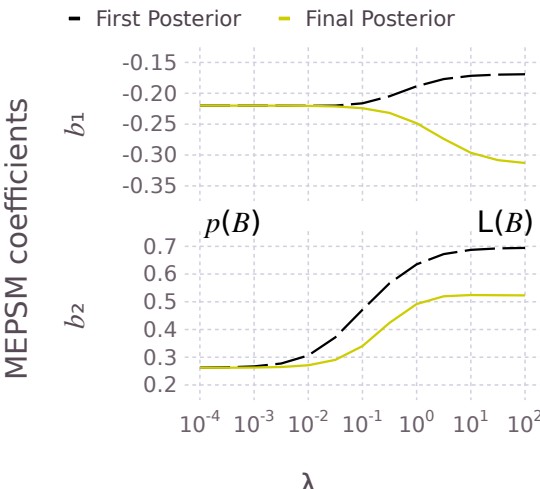

**Figure 3.** The effect of the regularization parameter $\lambda$ on the SST ($b_1$, top) and SSS ($b_2$, bottom) coefficients. The left and right side of the plots correspond to the prior distribution $p(\boldsymbol{B})$ and likelihood distribution $L(\boldsymbol{B})$ respectively.

Another advantage of Bayesian analysis, is that attaching a ridge parameter $\lambda$ to the prior variances means that the method can be used to regularize the solution, in a way that is different than traditional methods of regularization, such as prinicipal

components analysis, where components are retained by hard thresholding. Here, the ridge parameter $\lambda$ is attached to the prior:

$$p(B) = \mathcal{N}\left(\begin{bmatrix} b_0 \\ -0.22 \\ 0.27s \end{bmatrix}, \begin{bmatrix} 10^3 \sigma_{b_0}^2 & 0 & 0 \\ 0 & \lambda 0.02^2 & 0 \\ 0 & 0 & \lambda 0.15^2 \end{bmatrix}\right), \tag{26}$$

and allowed to vary over the range $10^{-4}$ to $10^2$. Figure 3 shows the effect of $\lambda$ for both the first and final posterior. As $\lambda$ increases, then $\boldsymbol{B}$ moves from the prior distribution to the likelihood distribution. The difference between the solid and dashed lines (for each coefficient) is because of the effect of adjusting $\Sigma_A$ for auto- and cross-correlation, which becomes less relevant

for small $\lambda$ ($\lambda < 10^{-2}$). More will be said about serial dependence in **A** below. This type of prior regularization will likely be useful for sites where the $(p+1)$-dimensional scatter of data (predictors plus response) is more spherical than ellipsoid e.g. a more spherical scatter of data could oocur at sites where the SST range is small. In practice, the value of $\lambda$ could be estimated using cross-validation methods.

Table 2 compares the regression coefficients (and variance) from 4 different solutions: a ML solution (Appendix F), the

WTLS method (Appendix C and D), and the MEPSM solution for the first and final posterior. The MLE (maximum likelihood estimation) solution and WTLS solution are basic implementations i.e. no prior, and no adjustment for autocorrelation or cross-





**Table 2.** Regression parameters and their standard deviation, from 4 different solutions

| Method | $b_0$ | $b_1$ | $b_2$ | Equations |
|---|---|---|---|---|
| MLE | -24.6±1.2 | -0.169±0.014 | 0.696±0.027 | Eq. (F3) |
| WTLS | -24.6±1.8 | -0.169±0.014 | 0.696±0.046 | Eq. (C8) & (D3) |
| First Posterior | -22.0±1.6 | -0.189±0.011 | 0.635±0.042 | Eq. (15) & (16) |
| Final Posterior | -15.4±2.2 | -0.248±0.017 | 0.492±0.061 | Eq. (15) & (16) |

correlation. The MLE and WTLS solutions are similar, with the variance being generally more conservative for WTLS, owing either to assumptions in the two methods, or differences in method implementation e.g. optimization for MLE versus analytical solution (with gaussian assumptions) for WTLS. For the first posterior of MEPSM, both the mean and the variance move away

from the WTLS solution, and towards a tight prior: in Fig. 3 the MEPSM first posterior (on the dashed line) is the same as the $\lambda = 10^0$ solution, and the basic WTLS solution is the same as $\lambda \geq 10^2$. For the final posterior (Table 2), the SST and SSS coefficients change steepness: $b_1$ becomes steeper, while $b_2$ becomes a little less steep compared to the WTLS solution and the first posterior (also seen on Fig. 3 as the difference between the solid and dashed lines). For both coefficients (Table 2), the variance increases for the final posterior (relative to the first posterior), because the final posterior adjusts for the auto- and

cross-correlation in $\Sigma_A$.

Figure 4 shows the autocorrelation of the noise in each variable ($\mathbf{A}_1$ is the SST noise, and $\mathbf{A}_2$ is the SSS noise), for the first and final posterior distribution. For the first posterior solution, the noise covariance in $\Sigma_A$ and $\Omega_V$ is assumed to be heteroscedastic but white. This assumption is clearly not true, because all variables appear to have seasonally-varying noise as shown by their autocorrelation functions (Fig. 4, left column). However, when this seasonally-varying noise is included in $\Sigma_A$

(but not in $\Omega_V$), then the response noise $V$ essentially becomes white, as all the seasonal dependence moves into the predictor noise (Fig. 4, right column). Whether or not the response noise $V$ should have seasonal variation (see Sect. 3.1) requires expert knowledge. This example is simply to illustrate the effect of including serially-correlated noise in a multipredictor model. The implications of seasonally-varying noise for coral PSMs will be addressed in another paper, by the application of MEPSM to a large coral database (Walter et al., 2022).

**4 Conclusions**

MEPSM is a new type of proxy system model which incoporates both prior information and generalized noise in all variables, such as cross-correlated noise within the predictor variables. In the work presented here, the response noise is assumed to be independent of the predictor noise (the zero matrices in $\Sigma_Q$), but cross-correlation between the predictor and response noise is possible (if needed), by complete quadratic multiplication of $\Sigma_Q$, rather than treating $\Omega_V$ and $\Sigma_A$ separately, in Eq. 8. However,

for many PSMs the predictor and response noise are fundamentally different, and therefore should be independent. The next



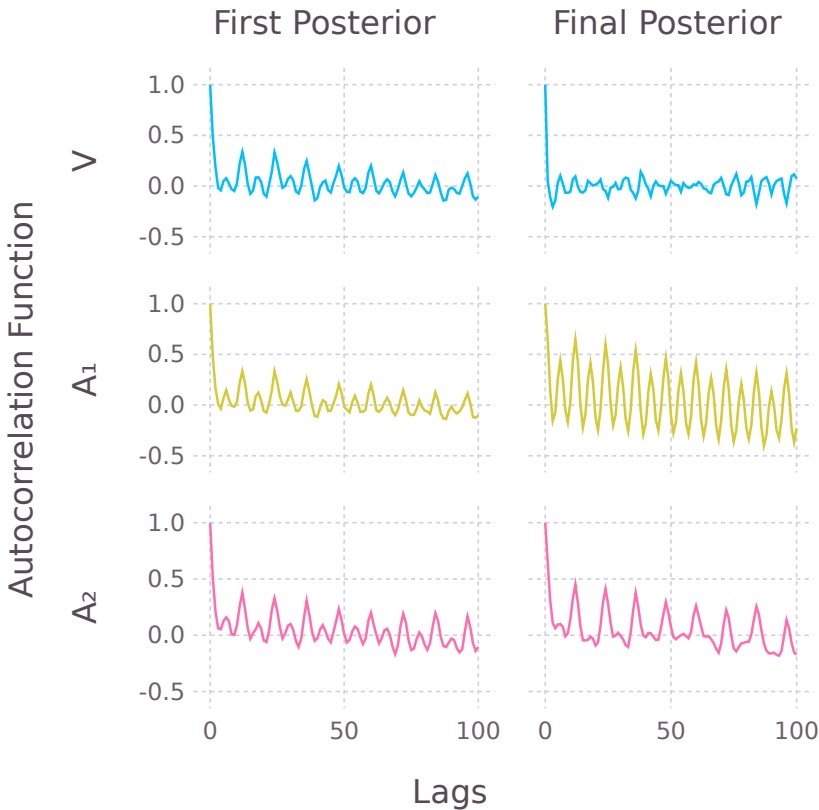

**Figure 4.** The autocorrelation function of the response noise ($V$), the SST noise $\mathbf{A}_1$, and the SSS noise $\mathbf{A}_2$, for the first posterior (left column), and the final posterior (right column). The predictor and response noise were calculated Equations.

step is to apply MEPSM to calibrating multiple-input proxy system models for different climate proxies, e.g. isotopes in corals, tree-rings, speleothems, and to incorporate MEPSM into current data assimilation projects.

*Code availability.* Easy to read Julia code for MEPSM is available on Github (https://github.com/Mattriks/MeasurementErrorModels.jl), and Zenodo (Fischer, 2023). A manual for MEPSM is available at https://mattriks.github.io/MeasurementErrorModels.jl/dev/. The manual

contains 3 examples which reproduce results from this paper.

*Data availability.* The ERSSTv5 sea surface temperature data are available at https://psl.noaa.gov/data/gridded/data.noaa.ersst.v5.html (metadata: https://doi.org/10.7289/V5T72FNM). The ERSSTv5 uncertainty data were obtained from Boyin Huang, NOAA (22/10/2020).



The HadEN4 ocean salinity data (and uncertainty) are available at http://apdrc.soest.hawaii.edu/las/v6/dataset?catitem=16640. These data
are from version EN.4.2.1, and are the G10 analyses i.e. with the corrections of Gouretski and Reseghetti (2010).

The coral $\delta^{18}$O data from core RI6 were extracted from the CoralHydro2k database (Walter et al., 2022).

A Jupyter notebook (as a pdf) showing the preprocessing of these data is available in the Supplementary Material. The dataset stored for the
code examples in the github repository contains all 3 detrended variables (detrended without removing the intercept).

## Appendix A: Basis of Eq. (2)

Let

$$y = f(x_1 - a_1, \ldots, x_p - a_p). \tag{A1}$$

A first-order Taylor expansion gives:

$$y \approx f(x_1, \ldots, x_p) - \sum_{k=1}^{p} \left[ \frac{\partial f(x_1, \ldots, x_p)}{\partial x_k} a_k \right], \tag{A2}$$

such that the variance of $f(\boldsymbol{x})$, $\mathrm{Var}(f(\boldsymbol{x})) \approx df(\boldsymbol{x})' \Sigma_{aa} df(\boldsymbol{x})$. The first-order Taylor exapnsion will be exact when $f(\boldsymbol{x})$ is a
linear function.


## Appendix B: Notation for covariance

Covariance can be given in plain or scaled format, e.g.

$$\Sigma = \sigma^2 \mathbf{S} \tag{B1}$$

So, just as an example, a naive covariance matrix for $\boldsymbol{B}$ could be written:

$$\Sigma_B = (\mathbf{X}^* \Sigma^{-1} \mathbf{X}^*)^{-1} = \sigma^2 (\mathbf{X}^* \mathbf{S}^{-1} \mathbf{X}^*)^{-1} \tag{B2}$$

In this paper I use the plain format because it simplifies many expressions.

## Appendix C: The WTLS likelihood, and its mean $B_{\text{wtls}}$

The target function for WTLS essentially consists of the quadratic component of the loglikelihood function:

$$\Phi_{\text{wtls}}(\boldsymbol{B}, \mathbf{X}^*) = \left( \mathrm{vec}(\mathbf{Z}) - \begin{bmatrix} \boldsymbol{Y}^* \\ \mathrm{vec}(\mathbf{X}^*) \end{bmatrix} \right)' \Sigma_Q^{-1} \left( \mathrm{vec}(\mathbf{Z}) - \begin{bmatrix} \boldsymbol{Y}^* \\ \mathrm{vec}(\mathbf{X}^*) \end{bmatrix} \right) \tag{C1}$$

where $\boldsymbol{Y}^* = \begin{bmatrix} \mathbf{1}_n & \mathbf{X}^* \end{bmatrix} \boldsymbol{B}$. For $\partial \Phi / \partial \boldsymbol{B}$, Eq. (C1) reduces to:

$$\frac{\partial \Phi_{\text{wtls}}}{\partial \boldsymbol{B}} = \frac{\partial [(Y - b_0 - X^* B^*)' \Omega_V^{-1} (Y - b_0 - X^* B^*)]}{\partial \boldsymbol{B}} \tag{C2a}$$



$$= 2\tilde{\mathbf{X}}'\Omega_V^{-1}\boldsymbol{V} \tag{C2b}$$

where $\tilde{\mathbf{X}} = \begin{bmatrix} \mathbf{1}_n & \mathbf{X}^* \end{bmatrix}$. Next substitute $\boldsymbol{V}$ with Eq. 18:

$$\frac{\partial \Phi_{\text{wtls}}}{\partial \boldsymbol{B}} = 2\tilde{\mathbf{X}}'\mathbf{W}^{-1}\boldsymbol{e} \tag{C3}$$

The residual vector $e$ can be written as:

$$\boldsymbol{e} = (\boldsymbol{Y} - b_0 - \mathbf{A}\boldsymbol{B}^*) - \mathbf{X}^*\boldsymbol{B}^*, \quad \text{or} \tag{C4}$$

$$\boldsymbol{e} = \boldsymbol{Y} - \mathbf{X}\boldsymbol{B} \tag{C5}$$

Then setting Eq. (C3) to zero, substituting for $\boldsymbol{e}$, and rearranging gives:

$$\boldsymbol{B}_{\text{wtls}} = \left[(\mathbf{X} - \mathbf{A})'\mathbf{W}^{-1}(\mathbf{X} - \mathbf{A})\right]^{-1}\left[(\mathbf{X} - \mathbf{A})'\mathbf{W}^{-1}(\boldsymbol{Y} - \mathbf{A}\boldsymbol{B})\right], \quad \text{or} \tag{C6}$$

$$\boldsymbol{B}_{\text{wtls}} = \left[(\mathbf{X} - \mathbf{A})'\mathbf{W}^{-1}\mathbf{X}\right]^{-1}\left[(\mathbf{X} - \mathbf{A})\mathbf{W}^{-1}\boldsymbol{Y}\right] \tag{C7}$$

where $\mathbf{A}$ now may include a column of zeros (corresponding to noiseless predictors in $\mathbf{X}$).

Eq. (C7) expands to:

$$\boldsymbol{B}_{\text{wtls}} = \left[\mathbf{X}'\mathbf{W}^{-1}\mathbf{X} - \mathbf{A}'\mathbf{W}^{-1}\mathbf{X}\right]^{-1}\left[\mathbf{X}'\mathbf{W}^{-1}\boldsymbol{Y} - \mathbf{A}'\mathbf{W}^{-1}\boldsymbol{Y}\right] \tag{C8}$$

where $(\mathbf{A}'\mathbf{W}^{-1}\mathbf{X})$ is the weighted covariance between $\mathbf{A}$ and the predictors, and $(\mathbf{A}'\mathbf{W}^{-1}\boldsymbol{Y})$ is the weighted covariance

between between $\mathbf{A}$ and the response vector. This shows that WTLS can be expressed as an adjusted least squares problem. Hence, it can also be expressed as an augmented linear regression:

$$\boldsymbol{Y} - \mathbf{A}\boldsymbol{B} = (\mathbf{X} - \mathbf{A})\boldsymbol{B} + (\boldsymbol{V} - \mathbf{A}\boldsymbol{B}) \tag{C9}$$

which is equivalent to Eq. (6)

## Appendix D: Covariance of $B_{\text{wtls}}$


One approximation for $\Sigma_B$ is as follows. Let

$$\boldsymbol{B} = \mathbf{K}\boldsymbol{Y} = \mathbf{K}(\mathbf{X}\boldsymbol{B} + \boldsymbol{e}) \tag{D1}$$

If $\mathbf{K}\mathbf{X} = \mathbb{I}$, then

$$\Sigma_B = \mathbf{K}\Sigma\mathbf{K}', \tag{D2}$$

where $\Sigma = \text{Diag}(\boldsymbol{e} \otimes \boldsymbol{e})$. For $\boldsymbol{B}_{\text{wtls}}$, from Eq. (D1) and Eq. (C7) it follows that $\mathbf{K} = \left[(\mathbf{X} - \mathbf{A})'\mathbf{W}^{-1}\mathbf{X}\right]^{-1}(\mathbf{X} - \mathbf{A})\mathbf{W}^{-1}$.
Therefore,

$$\Sigma_B^{\text{wtls}} = \left[(\mathbf{X} - \mathbf{A})'\mathbf{W}^{-1}\mathbf{X}\right]^{-1}\mathbf{M}\left[(\mathbf{X} - \mathbf{A})'\mathbf{W}^{-1}\mathbf{X}\right]^{-1'} \tag{D3}$$





where $\mathbf{M} = (\mathbf{X} - \mathbf{A})' \mathbf{W}^{-1} \Sigma \mathbf{W}^{-1} (\mathbf{X} - \mathbf{A})$

**Appendix E: Construction of $\Sigma_A$ and $\Omega_V$**

For two predictors, $\Sigma_A$ is:

$$\Sigma_A = \begin{bmatrix} \Omega_{A_1} & \Omega_{A_1 A_2} \\ \Omega_{A_2 A_1} & \Omega_{A_2} \end{bmatrix} \tag{E1}$$

$\Sigma_A$ can be constructed as:

$$\Sigma_A = \mathrm{Diag}(\sigma_A)' \begin{bmatrix} \mathbf{R}_{A_1} & \mathbf{R}_{A_1 A_2} \\ \mathbf{R}_{A_2 A_1} & \mathbf{R}_{A_2} \end{bmatrix} \mathrm{Diag}(\sigma_A) \tag{E2}$$

where $\mathrm{Diag}(\cdot)$ is a diagonal matrix, and $\sigma_A$ is a $(n \cdot p)$-length vector containing the hetereoscedastic "variance" of the predictor noise for each predictor (stacked vertically). The central block matrix, made up of submatrices $\mathbf{R}$, will be labelled $\widetilde{\mathbf{R}}$. The submatrices of $\widetilde{\mathbf{R}}$ are $n \times n$ correlation matrices, containing the autocorrelation or cross-correlation information for the predictor noise $\mathbf{A}$:

$$\mathbf{R} = \begin{bmatrix} \rho_0 & \rho_1 & \rho_2 & \rho_3 & \cdots & \rho_{n-1} \\ \rho_1 & \rho_0 & \rho_1 & \rho_2 & \cdots & \rho_{n-2} \\ \rho_2 & \rho_1 & \rho_0 & \rho_1 & \cdots & \rho_{n-3} \\ \rho_3 & \rho_2 & \rho_1 & \rho_0 & \cdots & \rho_{n-3} \\ \vdots & \vdots & \vdots & \vdots & \ddots & \vdots \\ \rho_{n-1} & \rho_{n-2} & \rho_{n-3} & \rho_{n-4} & \cdots & \rho_0 \end{bmatrix} \tag{E3}$$

where $(\rho_0, \rho_1, \ldots, \rho_{n-1})$ is the auto- or cross-correlation function.

If $\mathbf{R}$ is an autocorrelation matrix then $\rho_0 = 1$. Also, because the auto- or cross-correlation may become $\sim 0$ at long lags, then $\rho_{>k}$ may be set to 0, for a chosen lag $k$. To ensure that $\widetilde{\mathbf{R}}$ is positive definite, the algorithm of Rebonato and Jaeckel (1999) is computationally fast.

$\Omega_V$ can be similarly constructed:

$\Omega_V = \mathrm{Diag}(\sigma_V)' \mathbf{R}_V \mathrm{Diag}(\sigma_V)$ (E4)



**Appendix F: Maximum likelihood solution of Hannart et al. (2014)**

Hannart et al. (2014) defined the loglikelihood of the EIV model as (here written using the notation in this paper, rather than the notation of Hannart et al. (2014)):

$$\ell(\boldsymbol{B}^*, \mathbf{X}^*) = (\boldsymbol{Y} - b_0 - \mathbf{X}^* \boldsymbol{B}^*)' \Omega_V^{-1} (\boldsymbol{Y} - b_0 - \mathbf{X}^* \boldsymbol{B}^*) + \sum_i (\mathbf{X}_i - \mathbf{X}_i^*)' \Omega_{A_i}^{-1} (\mathbf{X}_i - \mathbf{X}_i^*) \tag{F1}$$

which is a reduced form of Eq. (C1), because the off-diagonal matrices $\Omega_{A_i A_j}$ in $\Sigma_A$ are not included. Hannart et al. wrote an iterative (fixed-point) solution in terms of $\mathbf{X}^*$ and $\boldsymbol{B}^*$.

$$\mathbf{X}_i^* = (\Omega_{A_i}^{-1} + b_i^2 \Omega_V^{-1})^{-1} (b_i \Omega_V^{-1} \bar{\boldsymbol{y}}_i + \Omega_{A_i}^{-1} \mathbf{X}_i) \tag{F2a}$$

$$\boldsymbol{B}^* = (\mathbf{X}^{*'} \Omega_V^{-1} \mathbf{X}^*)^{-1} \mathbf{X}^{*'} \Omega_V^{-1} (\boldsymbol{Y} - b_0) \tag{F2b}$$

where here $\boldsymbol{B}^* = [b_1 \ b_2]'$, and $\bar{\boldsymbol{y}}_i = \boldsymbol{Y} - b_0 - \mathbf{X}_{-i}^* \boldsymbol{B}_{-i}^*$. (I have included the intercept $b_0$ explicitly.) Confidence intervals

for each $b_i$ were calculated from the likelihood profiles (see Sect. 3 in Hannart et al., 2014).

The ML solution I provide here differs from Hannart et al. (2014), but I adopt a similar form (what follows is only for the ML estimate in the first row of Table 2). I express the loglikelihood in Eq. F1 as a function of $\boldsymbol{V}$ amd $\mathbf{A}$:

$$\ell(\boldsymbol{B}) = \boldsymbol{V}' \Omega_V^{-1} \boldsymbol{V} + \sum_i \mathbf{A}_i \Omega_{A_i}^{-1} \mathbf{A}_i \tag{F3}$$

where $\mathbf{A}$ and $\boldsymbol{V}$ are calculated using Eqs. 17-18. The difference with Hannart et al. (2014), is that the likelihood here is

rewritten with respect to $\boldsymbol{e} = \boldsymbol{Y} - \mathbf{X} \boldsymbol{B}$ (through Eqs. 17-18), whereas Hannart et al.'s procedure is expressed with respect to $\boldsymbol{Y} - \mathbf{X}^* \boldsymbol{B}^*$ i.e. the main difference being $\mathbf{X}$ or $\mathbf{X}^*$. The point here is merely to provide a basic ML solution in Table 2, in order to compare with the other solutions. Equation (F3) should work with many standard ML packages, because there is no iterative dependence on $\mathbf{X}^*$.

The first row in Table 2 was calculated using Eq. (F3), and the Julia package ProfileLikelihoods.jl (VandenHeuvel, 2022).

*Author contributions.* This paper is the work of MF.

*Competing interests.* The author declares that he has no conflict of interest.

*Acknowledgements.* Acknowledge Reviewers



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
