# Peer review of "Measurement Error Proxy System Models: MEPSM v0.2"

_EGUsphere, 2023_

## Author Comment (AC1)

**Response to Reviewer 2**

April 30, 2024

**Overview**

Proxy system models (PSMs) translate climate signals to proxy measurements and make the direct comparison between climate model simulations and proxy observations possible. They are also known as "forward data models" in contrast to "inverse models" that translate proxy measurements to climate signals. Since the forward process is natural (climate affects proxy formation; not the other direction), and the solution of an inverse process could be non-unique, PSMs have become a key component in paleoclimate data assimilation frameworks that bridges climate model simulations and proxy observations.

Existing statistical PSMs usually yield output with uncertainties, assuming the input climate signals are noiseless. This study proposes a new PSM framework that considers noises not only in the output, but also in the input. The comparison against existing PSMs may improve our understanding of the impact of the input noise on PDA reconstructions. I find the topic of this study to be important, and the proposed approach is noteworthy. The manuscript overall maintains a fine quality, although there are instances where the writing seems informal, and several places require clarification regarding the details. Furthermore, a major concern of mine is that while an application example is presented and some differences are compared, it is still unclear, at least not straightforward enough, on how the new PSM framework shows improvement compared to the traditional ones in terms of proxy data modeling accuracy. Therefore, I think a major revision is necessary. A few specific comments are listed below.

I thank the reviewer for their comments, and for the opportunity to expand on the discussion of MEPSM. The reviewer uses words like "accuracy" and "improvement", but in DA we should look at this wrt both the analysis mean state $\bar{\theta}^a$ and the analysis error covariance $\Sigma^a$ (i.e. is the estimation of the covariance accurate?) Also understanding the effect of predictor uncertainty on *calibrating* a PSM, means that we can ask questions about how good is the predictor data itself in different regions (which is a topic of a separate study).

I've reordered the Specific comments below, for clarification reasons.

**Specific comments**

- L31: What does "a" represent here; noise? Also, the word "unobservable" requires some clarification to readers not familiar with the method, as it is used many times in the manuscript. Does it mean the real signal without noise?

  Yes in $\mathbf{x} = \mathbf{x}^* + \mathbf{a}$, $\mathbf{a}$ is the noise, and $\mathbf{x}^*$ (the unobservable) is the real signal without noise. For example, SST products are not noiseless (Kennedy, 2014), and hence the true SST is unobserved. Fortunately, some climate variable data products are now providing gridded estimates of timevarying uncertainty, so we should make use of it. I will clarify "unobservable" and "$\mathbf{a}$" (in the text and Table 1).

- The idea of introducing prior noises is understandable. However, the setup of the current PDA (paleodata assimilation) projects applies an ensemble prior that samples potential prior errors as the input for the traditional statistical PSMs, which eventually can take into account the impact of the prior noises. In this ensemble prior setting, does the proposed new PSM framework show significant advantage? More discussions are needed.

The reviewer is asking about the connections between MEPSM and paleodata assimilation. To discuss the differences, I will use the following notation:

$\mathbf{x}^b$ is the climate model data, at $m$ gridpoints

$\mathbf{y}$ is the proxy data, at $p$ gridpoints

$\theta$ is the unobserved state vector, with $m$ gridpoints

$\mathbf{H}$ is the operator which converts model space to proxy space, $\mathbf{H}$ is $p \times m$

(Note that if a proxy system model has multiple inputs, then $\mathbf{H}$ can be formed by concatenation: e.g. $\mathbf{H} = [\mathbf{H}_{sst} \quad \mathbf{H}_{sss}]$ in which case $m$ above becomes $2m$.)

Data assimilation can be expressed in GLS format:

$$\mathbf{y} = \mathbf{H}\theta + \mathbf{r} \tag{1a}$$

$$\mathbf{x}^b = \theta + \boldsymbol{\varepsilon} \tag{1b}$$

where $\mathbf{r} \sim \mathcal{N}(0, \Sigma_{rr})$ and $\boldsymbol{\varepsilon} \sim \mathcal{N}(0, \Sigma_{\varepsilon\varepsilon})$. In the typical offline PDA, $\Sigma_{\varepsilon\varepsilon}$ is a background covariance matrix. In MEPSM, the predictor noise matrix is $\Sigma_A$. Out of interest, a time-constant noise matrix could be specified as $\Sigma_A = \Sigma_{aa} \otimes \mathbb{I}_n$. Note the difference between $\Sigma_{\varepsilon\varepsilon}$ and $\Sigma_{aa}$: $\Sigma_{\varepsilon\varepsilon}$ is estimated from climate model ensembles, whereas $\Sigma_{aa}$ is from ensemble climate data reconstructions (e.g. an ensemble SST product). Also, PDA is about estimating $\theta$, whereas MEPSM is about calibrating $\mathbf{H}$ (in MEPSM $\mathbf{B}$ contains the non-zero elements of $\mathbf{H}$). Note that $\mathbf{H}$ is typically calibrated using observed modern climate data and proxies, over a coeval period e.g. 20th Century. I will say more about $\mathbf{H}$ below.

Next concatenate the two equations above into one:

$$\mathbf{z} = \begin{bmatrix} \mathbf{y} \\ \mathbf{x}^b \end{bmatrix} = \widetilde{\mathbf{H}}\theta + \mathbf{q}, \tag{2}$$

where $\widetilde{\mathbf{H}} = \begin{bmatrix} \mathbf{H} \\ \mathbb{I}_m \end{bmatrix}$, and let $\Sigma_{qq} = \begin{bmatrix} \Sigma_{rr} & 0 \\ 0 & \Sigma_{\varepsilon\varepsilon} \end{bmatrix}$. The GLS solution for $\theta$ is:

$$\begin{aligned} \theta^a &= \left( \widetilde{\mathbf{H}}' \Sigma_{qq}^{-1} \widetilde{\mathbf{H}} \right)^{-1} \widetilde{\mathbf{H}}' \Sigma_{qq}^{-1} \mathbf{z} \\ &= \left( \mathbf{H}' \Sigma_{rr}^{-1} \mathbf{H} + \Sigma_{\varepsilon\varepsilon}^{-1} \right)^{-1} \left( \mathbf{H}' \Sigma_{rr}^{-1} \mathbf{y} + \Sigma_{\varepsilon\varepsilon}^{-1} \mathbf{x}^b \right), \end{aligned} \tag{3}$$

and for the analysis error covariance $\Sigma^a$:

$$\Sigma^a = \left( \mathbf{H}' \Sigma_{rr}^{-1} \mathbf{H} + \Sigma_{\varepsilon\varepsilon}^{-1} \right)^{-1}. \tag{4}$$

Eq. 4 can also be expanded as:

$$\begin{aligned} \Sigma^a &= \Sigma^a (\Sigma^a)^{-1} \Sigma^a = \Sigma^a \left( \mathbf{H}' \Sigma_{rr}^{-1} \mathbf{H} + \Sigma_{\varepsilon\varepsilon}^{-1} \right) \Sigma^a \\ &= \mathbf{K} \Sigma_{rr} \mathbf{K} + \Sigma^a \Sigma_{\varepsilon\varepsilon}^{-1} \Sigma^a \end{aligned} \tag{5}$$

where $\mathbf{K} = \Sigma^a \mathbf{H}' \Sigma_{rr}^{-1}$. Now what happens if $\mathbf{H}$ was calibrated using a biased method e.g. OLS? Then looking at Eq. 4 it seems $\Sigma^a$ is also biased. The reviewer asks about ensemble filtering methods, which are used for their computational advantage: replacing a large $m \times m$ $\Sigma_{\varepsilon\varepsilon}$ by a smaller ensemble of model $\mathbf{x}^b$ vectors. The model $\mathbf{x}^b$ are filtered in a way that ensures the resulting ensemble of $\delta\theta^a$ (i.e. deviation from $\bar{\theta}^a$) agrees with Eq. 5. That doesn't help if $\mathbf{H}$ is biased. Eq. 1 here shows that the operator $\mathbf{H}$ is applied to the unobserved state $\theta$, not to the "observed" $\mathbf{x}^b$, so using OLS for calibrating $\mathbf{H}$ seems flawed. OLS calibrates a PSM on "observed" data, whereas MEPSM calibrates a PSM on the unobserved state ($\mathbf{x}^*$).

I can add the above explanation as an Appendix (because it is self-contained, but adds extra notation).

- Conclusions: While I appreciate the author's theoretical work, overall it seems to me that the manuscript lacks a convincing validation. The author states that the next step is to apply the proposed PSM to different proxy types and to incorporate it into PDA projects. However, in my opinion, more tests and validations, e.g., real-world data modeling tests on more sites, and perhaps even a pseudoproxy DA experiment if the author would like to make concrete connections to PDA applications, are actually needed for this study to clearly show that this new PSM indeed works. The analyses in Figs. 1-4 and Table 2 show only the differences, which may not necessarily be the improvements.

  I'm currently working on a separate manuscript where I apply MEPSM to the global CoralHydro2k database, using several marine datasets which provide uncertainty estimates. The manuscript asks questions such as "Are coral PSMs closer to the likelihood or the prior for MEPSM, and are there regional differences?". That question is also about how good are the 20th Century marine inputs for coral PSMs, and that cannot be properly answered without taking into account the uncertainty in the marine predictors, as in MEPSM. This type of analysis can be applied to other archives too e.g. speleothems, tree-rings etc (using appropriate predictor datasets). Following that, future papers are planned where MEPSM is incorporated into PDA projects. I think that the scope and the size of the forementioned CoralHydro2k-based mansucript (it currently has 11 main Figures) warrants a separate paper. I note the "Manuscript types" page (subsection Model description papers) for GMD says that "Where evaluation is very extensive, a separate paper focussed solely on this aspect may be submitted".

- - L32: "Appendix 1" $\rightarrow$ "Appendix A".
  - Code availability: "Easy to read" $\rightarrow$ "Intuitive"

  Corrected, thanks!

**References**

Kennedy, J. J.: A review of uncertainty in in situ measurements and data sets of sea surface temperature, Reviews of Geophysics, 52, 1–32, https://doi.org/10.1002/2013RG000434, 2014.

---

## Author Comment (AC2)

**Response to Reviewer 1**

May 19, 2024

**Overview**

This work proposes a proxy system model, which incorporates both prior information and generalized noise in all variables. This can be useful for paleoclimate data assimilation. The results look reasonable. The reviewer suggests this paper can be accepted after some minor revisions.

Thankyou for your comments.

1. I recommend the author to restructure Section 2.1, the existing methods/theories should be distinguished from the proposed methodology. The increment unique to this work needs to be clear. I've split Sect 2.1 now into two sections (2.1 and 2.2), and added Table 2 to make the historic advances clear. Table 2 is discussed in the new text in Section 2.1.

2. Because this paper has many symbols, I suggest all the symbols like those around Line 60 should be documented in Table 1, rather than being described in the text near the equations. Table 1 now contains most of the notation, including table footnotes with some explanatory text. I occasionally give extra information in the text below an equation where I think necessary. I also update some of the notation e.g. for $\theta$, $\Theta$ and $\phi$, $\Phi$.

3. On line 122, better use "Hadamard product" instead of "Hadamard multiplication". I've changed this. In the literature I do see Hadamard multiplication being used in an action word sense.

4. - Line 45, "exapands" $\rightarrow$ "expands"
   - Line 255, "incoporates" $\rightarrow$ "incorporates"
   Corrected, thanks!